# Physiological Modeling of Hemodynamic Responses to Sodium Nitroprusside

**DOI:** 10.3390/jpm13071101

**Published:** 2023-07-06

**Authors:** Joseph Rinehart, Sean Coeckelenbergh, Ishita Srivastava, Maxime Cannesson, Alexandre Joosten

**Affiliations:** 1Department of Anesthesiology & Perioperative Care, University of California Irvine, Orange, CA 92868, USA; 2Outcomes Research Consortium, Cleveland, OH 44195, USA; 3Department of Anesthesiology, Erasme Hospital, Université Libre de Bruxelles, 1050 Brussels, Belgium; 4Department of Anesthesiology and Intensive Care, Paul Brousse Hospital, Hôpitaux Universitaires Paris-Sud, Université Paris-Saclay, Assistance Publique Hôpitaux de Paris (APHP), Villejuif, 44195 Paris, France; 5Departments of Anesthesiology and Perioperative Medicine, David Geffen School of Medicine, University of California, Los Angeles, CA 90095, USA; mcannesson@mednet.ucla.edu

**Keywords:** hemodynamic modeling, sodium nitroprusside, lumped-parameter model

## Abstract

Background: Computational modeling of physiology has become a routine element in the development, evaluation, and safety testing of many types of medical devices. Members of the Food and Drug Administration have recently published a manuscript detailing the development, validation, and sensitivity testing of a computational model for blood volume, cardiac stroke volume, and blood pressure, noting that such a model might be useful in the development of closed-loop fluid administration systems. In the present study, we have expanded on this model to include the pharmacologic effect of sodium nitroprusside and calibrated the model against our previous experimental animal model data. Methods: Beginning with the model elements in the original publication, we added six new parameters to control the effect of sodium nitroprusside: two for the onset time and clearance rates, two for the stroke volume effect (which includes venodilation as a “hidden” element), and two for the direct effect on arterial blood pressure. Using this new model, we then calibrated the predictive performance against previously collected animal study data using nitroprusside infusions to simulate shock with the primary emphasis on MAP. Root-mean-squared error (RMSE) was calculated, and the performance was compared to the performance of the model in the original study. Results: RMSE of model-predicted MAP to actual MAP was lower than that reported in the original model, but higher for SV and CO. The individually fit models showed lower RMSE than using the population average values for parameters, suggesting the fitting process was effective in identifying improved parameters. Use of partially fit models after removal of the lowest variance population parameters showed a very minor decrement in improvement over the fully fit models. Conclusion: The new model added the clinical effects of SNP and was successfully calibrated against experimental data with an RMSE of <10% for mean arterial pressure. Model-predicted MAP showed an error similar to that seen in the original base model when using fluid shifts, heart rate, and drug dose as model inputs.

## 1. Introduction

Computational modeling of physiology is commonly used in both the early development and validation and safety testing of medical devices [1,2]. The benefits are myriad and include relatively low cost compared to that of in vivo and clinical trials, the ability to mitigate risk prior to introduction into clinical environments, and the ability to rapidly modify and iterate testing conditions during development [3,4,5]. Moreover, computational physiology models allow for testing in conditions or at extremes that may not be observed even in large clinical trials but are nevertheless necessary for safety assurance [6,7]. These models, and the credibility assessment that supports them, may be an essential element of regulatory submissions to agencies like the US Food and Drug Administration (FDA) or European Notified Bodies to support a marketing application [5,8,9,10].

Bighamian et al. have published a lumped-parameter computational model of core cardiovascular physiology that includes a fluid-shift mechanism (accounting for both intravascular losses like bleeding and urine production, and gains like crystalloid administration and redistribution), a stroke-volume mechanism, and a mean arterial pressure mechanism [11,12]. The authors (1) provide a foundational basis for the model elements; (2) validate individually parameterized implementations against experimental animal data for blood volume, mean arterial pressure (MAP), stroke volume (SV), and cardiac output (CO); and (3) perform an analysis of parameter sensitivity and performance when low-sensitivity elements are removed from the model. The strengths of this model include physiological and physics-based mechanisms, a well-considered fluid-shift mechanism describing not only blood volume changes but the differential responses to crystalloid and colloid administration and compensatory physiologic mechanisms, the ability to fully individualize models to specific subjects, validation against in vivo data, and parameter sensitivity analysis and subsequent model conditioning. Limitations noted by the authors of the model are that heart rate (HR) is assumed to be available as an input parameter (and the model is therefore not self-sufficiently capable of generating all physiologic parameters on its own from volume changes/parameters), the model does not explicitly consider the effects of the unstressed blood volume, and the model includes some non-linear elements that might complicate identification and observation of model outputs.

The goal of the present project was to begin extending this model to include effects of common pharmacologic agents used in anesthesiology and critical care in order to create a simulation model suitable for initial development of additional closed-loop drug controllers. We have previously performed a detailed in vivo experiment using a porcine model of hypotension induced by sodium nitroprusside (SNP) in the development of our closed-loop vasopressor system, so this was a natural place to start expanding the model [13,14,15,16]. Detailed physiological data including cardiac stroke volume and precise times for initiation and changes in the SNP infusion were recorded, providing a robust data source for calibration of the model and identification of appropriate parameter ranges and/or simulation of individuals. With a focus on potential use in the development of future closed-loop pharmacologic interventions [17,18,19,20], we hypothesized that the present model could be extended to include the effects of SNP with a root-mean-squared error of less than 10% for MAP itself, as was achieved with the original model.

## 2. Methods

The present study was performed completely in silico and was therefore non-human-subjects research. The animal data used in this study was collected previously at Erasme University Hospital in Brussels, Belgium, and that study was approved by the Institutional Animal Ethics Committee on 8 February 2018 (LA1230336) [13].

### 2.1. Base Computational Model

All of the computational model and analysis work was done in Python 3.9 (www.python.org) (accessed on 3 July 2023).

For the base computational model we began with an implementation of the model described by Bighamian et al. [11]. This is a lumped-parameter model and includes three sub models: fluid-shifts in response to blood volume perturbations from bleeding, fluid administration, and renal excretion; a model for relating blood volume to cardiac stroke volume and cardiac output; and a model for relating cardiac output to blood pressure. 

For our purposes, the model was implemented using the author’s final Equations (5), (12) and (14) as described in the original manuscript and shown below. For any timepoint *t*, fluid shift, cardiac output (*CO*), and mean arterial blood pressure (*BP*), respectively, can be calculated as:(5)$${\Delta \dddot{V}}_{B}\left(t\right)+K_{p}{\Delta \ddot{V}}_{B}\left(t\right)+K_{i}\Delta {\dot{V}}_{B}\left(t\right)=\left[\ddot{u}\left(t\right)-\ddot{v}\left(t\right)\right]+\frac{K_{p}}{1+A_{u}}\;\dot{u}\left(t\right)-\frac{K_{p}}{1+A_{v}}\;\dot{v}\left(t\right)+\frac{K_{i}}{1+A_{u}}\;u\left(t\right)-\frac{K_{i}}{1+A_{v}}\;v\left(t\right)$$
(12)$$CO\left(t\right)=HR\left(t\right)*SV\left(t\right)=HR\left(t\right)*\;\theta_{1}\log\left(\theta_{2}*CO\left(t\right)+\theta_{3}V_{B}\left(t\right)+\theta_{4}\right)$$
(14)$$BP\left(t\right)=CO\left(t\right)*\left({TPR}_{0}-\frac{\Delta TPR*\;sgn\left(BP\left(t\right)-{BP}_{0}\right)*\;\sqrt[{3}]{\left|BP\left(t\right)-{BP}_{0}\right|}}{2*\left(1+\sqrt[{3}]{\left|BP\left(t\right)-{BP}_{0}\right|}\right)}\right)$$
where Δ*V* represents the rate of change in blood volume with dots representing degrees of derivative; *K_p_* and *K_i_* are proportional and integral gains for the fluid-shift mechanism; *u* and *v* are model inputs describing the rates of blood volume/fluid gain and loss, respectively (with dots again representing the degree of derivative); *A_u_* and *A_v_* are parameters describing the ratio of steady-state intravascular to extravascular distribution of fluid gain and loss, respectively; thetas are subject-specific parameters defining stroke volume response to blood volume (*θ*_3_), current cardiac output (*θ*_2_), and general gain/curve parameters (*θ*_1_ and *θ*_4_); TPR_0_ is the total peripheral resistance at time zero calculated as:$${TPR}_{0}=\frac{{BP}_{0}}{{CO}_{0}}$$
and TPR at time *t* is calculated as the large parenthesis in Equation (14) in the previous time point. The authors additionally evaluated *K_p_* and *K_i_* independently in the original model for crystalloid and colloid, so we implemented them independently as separate mechanisms for the two different fluid types in this model.

### 2.2. Addition of SNP to the Model

The net effect of SNP on the cardiovascular system is arterial and venous venodilation, reduced afterload, decreased ventricular filling pressures, lower systemic blood pressure, and relatively unchanged heart rate [21]. In the Bighamian model, central venous pressure is not explicitly tracked, so venous dilation and decreased ventricular filling pressures would need to be accounted for in the stroke volume calculation itself (Equation (12)). Afterload and blood pressure could then separately be accounted for in the blood pressure equation (Equation (14)). In the original model, the fluid-shift mechanism (Equation (5)) was not dependent on any of the hemodynamics, so for the present work, no modifications were indicated or made. 

We chose to use the parameter phi (ϕ) to describe the pharmacologic elements of the six new parameter additions used to model SNP. For simplicity, we chose to model drug elimination as a first-order effect with a subject-specific half-life defined as ϕ_1_. For each second of elapsed simulated time, the amount of drug in the plasma was reduced using the following equation:$${NTG}_{plasma}={NTG}_{plasma}\left(1-\frac{0.5}{\phi_{1}}\right)$$

There is an intrinsic delay between the “injection” of a drug and the effects of that drug at the target receptor (in effect the, “onset time”). This delay will include infusion line delay, circulation time, and properties of the drug itself and the target receptors. These effects were collectively modeled as ϕ_2_. There is no specific equation this variable factors into, rather, any drug effect “added” to the system is placed into a queue that delays the effect for ϕ_2_ s. Finally, since SNP needs to affect mean arterial pressure directly, and then venous tone and thereby cardiac stroke volume, we defined ϕ_3_ and ϕ_4_ as the curve and gain parameters for the drug effect on arterial vascular tone, respectively, and ϕ_5_ and ϕ_6_ as the curve and gain effect of the drug on cardiac stroke volume (via the venodilation effect that is hidden).

To implement the arterial effect and in anticipation of the desire to incorporate additional interacting pharmacologic effects in the model in the future, we sought to create a single mechanism that could account for the combined influence of physiologic factors on arterial vascular tone. We chose the symbol α for this effect in reference to alpha-adrenergic receptors found in arterial vasculature and a priori defined the range of this parameter as (100,−100) representing the percentage of combined possible receptor activity for all vasodilatory and vasoconstrictive influences within the subject. With only SNP included in the present model, α is calculated as: $$\alpha =100\;*\;\phi_{4}\left(1-\frac{\phi_{3}}{\phi_{3}+SNP(t)}\right)$$
where ϕ_3_ is a subject-specific parameter that defines the SNP response curve inflection point, φ_4_ is a subject-specific parameter that defines the overall potency of the drug, and SNP(t) is the blood concentration of SNP in nanograms per milliliter. From this, the original Equation (14) was then modified to include α’s effect on systemic resistance (changes shown in bold):(14b)$$BP\left(t\right)=CO\left(t\right)*\left({TPR}_{0}-\frac{\Delta TPR\;*\;sgn\left(BP\left(t\right)-{BP}_{0}\right)\;*\;\sqrt[{3}]{\left|BP\left(t\right)-{BP}_{0}-\alpha \right|}}{2\;*\left(1+\sqrt[{3}]{\left|BP\left(t\right)-{BP}_{0}-\alpha \right|}\right)}\right)$$
where α is the percentage of combined possible receptor activity for all vasodilatory and vasoconstrictive influences within the subject, and TPR_0_ is the total peripheral resistance at baseline. TPR at time *t* is calculated as the value of the large parenthesis in Equation (14b), and ΔTPR is then calculated as the difference between TPR_0_ and TPR_t_ in the previous calculation.

Finally, in order to account for the drop in filling pressures from SNP, we added a fifth parameter and term to Equation (12) specifically linked to the SNP concentration (changes shown in bold):(12b)$$CO\left(t\right)=HR\left(t\right)\;*\;SV\left(t\right)=HR\left(t\right)\;*\;\theta_{1}\log\left(\theta_{2}\;*\;CO\left(t\right)+\theta_{3}V_{B}\left(t\right)+\theta_{4}+\phi_{5}(SNP(t)/(\phi_{6}+SNP\left(t\right))\right)\;$$
where *SNP*(*t*) is the circulating SNP concentration in nanograms per milliliter, ϕ_5_ is a subject-specific parameter reflecting the subject’s net stroke volume response sensitivity to the drug (following from the venodilation that is not explicitly tracked in the model), and ϕ_6_ is a parameter dictating the sharpness of the response curve. We initially attempted to model this effect using only a single parameter, but the stroke volume responses in some subjects showed high dose sensitivity where other animals exhibited a more on/off response; thus, this last parameter was ultimately necessary to determine how sensitive the response was to the dose versus the overall total effect.

### 2.3. Animal Data

The calibration data used in this study was taken from a previous study performed by the authors as noted above [13]. In that protocol, a total of 16 pigs were studied, 14 of which were randomized into closed-loop treatment or no treatment (the control). Only the 7 control animals from that study were used in the present work. The key points of that protocol relevant to the present project are summarized here.

The animals were fasted, anesthetized, and monitoring was placed as described in the original protocol, including advanced hemodynamic monitoring (EV-1000, Edwards Lifesciences, Irvine, CA, USA) for cardiac stroke volume. All animals then underwent a two-hour study protocol during which four hypotensive phases (30 min each) were induced by fixed SNP infusion rates (doses between 65 and 130 µg/min). First, an SNP infusion was initiated and increased until 130 µg/min, the dose that had caused a reduction in MAP to around 50 mmHg in the two pilot animals. This infusion rate was continued for 30 min. The infusion rate was then decreased to 65 µg/min for 30 min. In the third phase, the SNP rate was increased again to 130 µg/min for 30 min, and finally, in the fourth phase, decreased by half again for the last 30 min of the study protocol. Hemodynamic variables were recorded for 10 min after the discontinuation of the SNP to evaluate the return to baseline. 

Fluid administration was standardized in all animals: 500 mL of crystalloid was administered during induction and then a 5 mL/kg/h infusion run. Additional 100 mL boluses of 6% hydroxyethyl starch were administered when recommended by the Assisted Fluid Management decision support on the monitor (Hemosphere, Edwards Lifesciences, Irvine, CA, USA). Bolus start and stop times were recorded in the original protocol, making it possible to accurately reproduce the fluid administrations for our modeling purposes.

### 2.4. Individual Model Fitting

Using this animal data, our aim was to evaluate whether the modeling of SNP introduced by our modifications to the base hemodynamic model could be fit to individual hemodynamic responses after individualized fitting of the model parameters to the subject. The present model adds a total of five new parameters to the original Bighamian model related to SNP. Parameter identification complexity can increase non-linearly with the number of parameters depending on their interactions, so for the present project, since fluid response was not the primary topic of interest, the fluid-related values (A_u_, A_v_, K_p_, and K_i_) were standardized as shown in Table 1 to the population median from the initial study to reduce complexity in fitting the new parameters.

The original study data had core hemodynamics (HR, MAP) recorded in 2 s intervals and advanced parameters recorded in 20 s intervals (SV, CO). In preparation for this work, the original data was first reduced to 20 s intervals for all hemodynamics. For each subject, fluid administrations were then added to the data set as total milliliters given, discriminating between crystalloid and colloid, and then the SNP infusion rate (recorded as micrograms given per interval) was added using the timings recorded from the original study. Next, unadjusted blood volume was calculated at each time point using the raw crystalloid and colloid inputs, and then the fluid-shift adjusted blood volume was calculated using Equation (5) above and parameters in Table 1.

Following these calculations, a grid-search fitting process was then performed to identify optimal parameters for each subject’s data. A grid-search is not a particularly robust optimization method and may mask an ill-conditioned problem, but it is computationally easy to implement and understand and despite the limitations and some of the arbitrary choices in implementation, it was used as an initial exploration of the feasibility of this approach. The necessary caveat to this approach is that better optimization will be indicated in future work alongside true validation. The grid-search was conducted as follows:First, using the initial starting parameters shown in Table 1, a plasma SNP concentration was calculated from the administration rates, φ_1_ (half-life), and φ_2_ (infusion-to-onset delay).Cardiac stroke volume was calculated using Equation (12b) above and the initial θ_1–4_ and φ_5_ parameters in Table 1.The root-mean-squared error (RMSE) was then calculated between the measured SV and the simulated SV. RMSE was used as the minimization criterion as this is the parameter reported by the authors in the original Bighamian model, so it made a useful direct comparator.Each of the parameters θ_1_, θ_2_, θ_3_, θ_4_, and φ_5_ was individually increased and decreased by 10%, and the cardiac stroke volume and resulting RMSE from the new set was recalculated. The modification that resulted in the largest decrease in RMSE was implemented.Step 4 was repeated until no modification of a parameter resulted in at least a 1% reduction in RMSE.φ_1_ and φ_2_ were then individually increased and decreased by 10% and the stroke volume and RMSE recalculated, and the process returned to step 1, calculating new plasma concentrations using the new values and then repeating the fitting process in steps 2–5. The change resulting in the largest reduction of RMSE for φ_1_ and φ_2_ was implemented.Step 6 was continued until no change in φ_1_ and φ_2_ parameters resulted in at least a 1% reduction in RMSE.Finally, once the process above was completed, since φ_3_ and φ_4_ affect only MAP, they were calculated last using a grid search process identical to steps 4 and 5 above but using a simulated MAP instead of a simulated SV (with said MAP calculated using the previously calculated simulated SV and the model parameters in Equation (14b)) against recorded MAP to calculate RMSE scores.

The final parameters from this model were then graphed for visual inspection and the simulated hemodynamics recorded along with the final RMSE.

Following the initial fitting, parameters with a coefficient of variation of ≤10% in the population would be fixed at their average values and the fitting process re-run to evaluate partially fit models.

### 2.5. Statistical Analysis and Reporting

Mean and SD for each fit parameter are reported along with the raw values for each subject. Root-mean-squared errors are reported for each comparison of experimental to predicted hemodynamics. Bland–Altman plots were used to calculate 95% limits of agreement between experimental and predicted hemodynamics.

## 3. Results

Table 1 shows the starting and individualized model parameters after fitting. Of the ten parameters fit in the model, three had coefficients of variation less than or equal to 10%: θ_1_, θ_3_, and φ_5_. These parameters were set to their population average values and fixed for the partially fit models. Table 2 shows the RMSE associated with the fully individualized models as well as the partially individualized models in reproducing SV, CO, and MAP. The individually fit models showed tighter fits than using the population average values did, suggesting that the fitting process was effective in identifying improved parameters. Use of the partially fit models after removal of the lowest variance population parameters showed a very minor decrement in improvement over the fully fit models (Table 2). Ultimately, the individually and partially fit models met the criteria for <10% RMSE in MAP, but the population models did not.

Figure 1 shows the graphs for the fully individualized models for SV, CO, and MAP.

Figure 2 shows Bland–Altman plots for the fully individualized model fits for SV, CO, and MAP. Bias was near zero for each, and 95% limits of agreement were ±23, ±2.0, and ±10, respectively.

## 4. Discussion

In the present project, a computational model for blood volume, stroke volume, and mean arterial pressure was expanded to include the pharmacologic effects of SNP and calibrated against previous animal data. Predictive ability error for MAP as measured by root-mean-squared error was <10% as desired at the outset, while the error values for SV and CO were higher at 12–15%. One notable limitation in the present study in that regard, however, is that the CO and SV were measured by a minimally invasive arterial line using a non-calibrated hemodynamic monitor (FloTrac transducer, Hemosphere monitor, Edwards Lifesciences, Irvine, USA). Minimally invasive and non-calibrated cardiac output monitoring is known to have up to 2 L-per-minute limits of agreement with invasive central monitoring methods (e.g., Swan-Ganz catheterization), and in particular was not developed for porcine physiology. Despite these known limitations, we felt it was important to include these parameters in the modeling as part of the overall model and considered a better alternative than simply abandoning any attempts at fidelity in flow-based prediction. Finally, the fixing of the crystalloid and fluid-shift mechanism parameters to population values instead of individual fitting as in the original work may have further reduced accuracy in SV and CO output.

Despite this limitation, we see excellent visual agreement with the prediction graphs in all evaluated hemodynamics, with a significant portion of the error coming from “faster dynamics” in the physiology data—minute-to-minute variations. The significance of missing these rapid dynamics may need to be evaluated further depending on the model applications; slow-control applications may be insensitive since the trends follow closely, but for rapid-control applications, they could be significant deviations in prediction of control performance from the true physiology. No data cleaning was performed (i.e., smoothing or moving averages) prior to comparing to the model outputs. There was a significant heart rate spike in subject 7 that was poorly accounted for in the CO model (note large uptick in predicted CO compared to drop in measured CO in bottom middle panel of Figure 1); this may indicate a need to revisit the heart-rate feedback mechanism in the original model, but as noted may also be a limitation of the cardiac output measurement modality. Mean arterial pressure also deviated significantly in this animal, however, further suggesting that the model may benefit from additional parameterization for high heart rates. Another possibility is that the SNP onset and/or metabolism times were too delayed, but manually reducing the onset time worsens error in all subjects.

We did not perform a parameter sensitivity analysis as extensive as was performed in the original model development. In that study, however, parameter θ_1_ was found to have low impact in the final model, which the present work supports, as it had low percent variation relative to that of other model parameters. Additionally, the initial results of the present study indicated that both θ_3_ and ϕ_5_ were low-variance model parameters and may be set at population averages with minimal impact on model output, which our partial model fittings supported.

As noted previously, the use of computational models to support medical research, device development, and safety testing is now routine. In all models there is a trade-off between model complexity and predictive capacity. The present model was able to expand on the original core fluid-shift and hemodynamics model with the addition of a response to SNP using six new parameters to define the drug response. If the partial model fit is used, only a net of three new parameters are needed relative to that in the original model. In modeling new pharmacologic agents one convenience is that when the agents are not present, the parameters reduce to zero and become silent in the model; this makes the addition of other agent effects easier in isolation. Complexity may increase rapidly as multiple simultaneous effects are modeled, however, and the experimental data necessary for validation may become more challenging to obtain. In the original animal model experiment from which the calibration data came, half of the animals received SNP alone as a control group, and the other half received norepinephrine (NE) titrated by a closed loop. With the SNP model in place, the NE-treated animals in that original study may serve as useful data for work with an NE model as a potential next step in the development of the present work. Additional directions may also include other vasodilators (nitroglycerine and nicardipine).

### 4.1. Limitations

One limitation is the use of minimally invasive cardiac output monitoring. As the primary interest of this model is MAP, however, and MAP error was reduced compared to that in the original work, the loss of accuracy in SV and CO may be allowable. Another potential limitation is that while the grid-search approach to model individualization was effective relative to using starting parameters or mean parameters for the cohort, it is possible that other mathematical optimization methods may be superior. A grid search may identify a local minimum, for example, where other approaches may be more robust. Moreover, the parameters in Table 1 have larger inter-subject variability, which suggests that there may be opportunity for improvement of the model, or perhaps additional parameterization may be of benefit. This will need to be explored in future work. The grid search has the benefit, however, of being readily understood and computationally straightforward. An additional limitation of the implementation of the pharmacologic effects of SNP is that ϕ_2_ includes several factors that contribute to onset delay: infusion line delay (the time it takes for a change in drug to be carried through the IV fluid to the subject), circulation time, and pharmacologic onset. While for the present study there is only one drug being evaluated, if more than one drug was included in the model, several of these features would need to be the same for all such drugs, and thus it might make sense to break this parameter out into individual components in a future model. Finally, the present model does not include specific central venous, right-sided cardiac, or pulmonary components. These could be added in a future expansion as we have done with previous work [22].

Despite the model’s limitations, it may still be useful (paraphrasing George Box’s famous adage). We have used relatively rudimentary and even unvalidated models (some mechanistic, some lumped parameter) in some of our other original work creating physiological sandboxes for development of closed-loop controllers [23,24], and what often occurs is a cyclical process of refinement of the controller alongside the model, where both slowly improve the other as additional data is collected and incorporated [25,26,27,28,29,30].

### 4.2. Conclusions

A previously validated computational model for fluid shift, cardiac output, and mean arterial pressure was expanded to include a response to SNP in the model. The model was calibrated against experimental data and showed lower root-mean-squared error for MAP compared to that of the original model at the cost of higher SV and CO error.

## Figures and Tables

**Figure 1 jpm-13-01101-f001:**
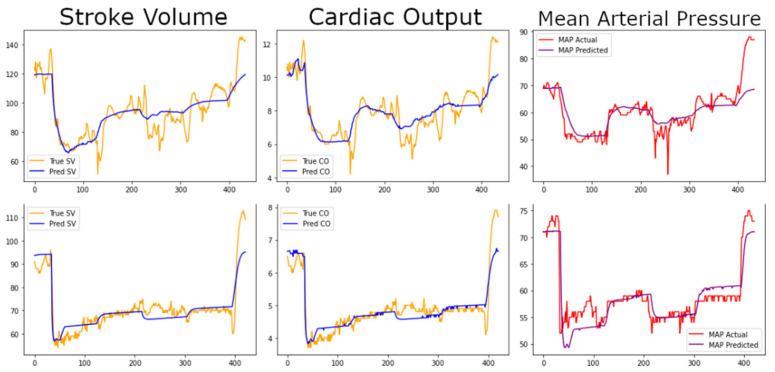
**Comparison plots of stroke volume, cardiac output, and mean arterial pressure in individually fit models.** Each row of plots represents a single validation subject’s data. In the stroke volume and cardiac output graphs, gold lines indicate the clinical data, and blue lines represent the simulation model data predicted from fluid volumes given, SNP concentration, and heart rate after individual fitting of model parameters for that subject. For the mean arterial pressure graphs, red lines indicate experimental data, and the purple line indicates the model-predicted values. CO—cardiac output; MAP—mean arterial pressure; SV—stroke volume.

**Figure 2 jpm-13-01101-f002:**
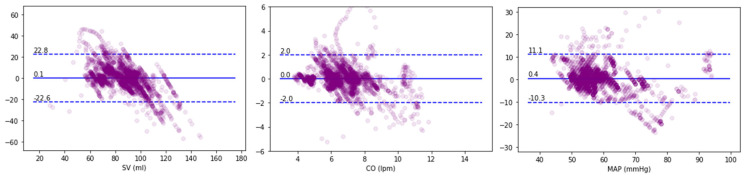
**Bland–Altman plots for error between model prediction and experimental data.** Bland–Altman analyses of predicted and experimental agreement for SV, CO, and MAP. Dotted blue lines mark the 95% confidence intervals for average difference, and the solid blue line—the average difference in measurement between predicted and experimental data. CO—cardiac output; MAP—mean arterial pressure; SV—stroke volume.

**Table 1 jpm-13-01101-t001:** Model parameters at initialization and after fitting.

Model Parameter	Initial Value	1	2	3	4	5	6	7	Mean	SD	% Var
A_u_ (crystalloid)	1.9	(fixed)							1.9	0	0
A_u_ (colloid)	0.0	(fixed)							0.0	0	0
Av	0.13	(fixed)							0.13	0	0
Kp	0.0031	(fixed)							0.0031	0	0
Ki	1.09	(fixed)							1.09	0	0
θ_1_	13	16	12	16	14	13	13	14	14	1.4	10
θ_2_	−1.0	−4.3	0.0	0.0	−6.9	−5.0	−4.6	−4.9	−3.7	2.6	−72
θ_3_	0.29	0.58	0.60	0.59	0.58	0.59	0.65	0.68	0.61	0.04	6
θ_4_	−420	46	0	0	0	101	130	175	65	71	111
φ_1_	120	144	46	187	207	108	120	120	133.2	53.2	40
φ_2_	60	104	72	179	149	60	60	60	98	49	50
φ_3_	200	1013	300	675	675	102	200	300	466.4	327.6	70
φ_4_	1.0	0.73	1.00	0.73	0.81	1.56	1.00	1.00	0.98	0.29	29
φ_5_	2000	2000	2208	1951	1951	1951	2000	2000	2009	91	5
φ_6_	5.0	21.1	1.9	7.0	63.5	19.2	14.7	17.3	20.7	20.1	97

A_u_, population standardized redistribution factor for fluid administration in the fluid-shift mechanism; A_v_, population standardized redistribution factor for volume loss in the fluid-shift mechanism; K_p_, proportional gain for the fluid-shift mechanism; K_i_, integral gain for the fluid-shift mechanism; θ_1_, general gain parameter; θ_2_, current cardiac output feedback parameter; θ_3_, stroke volume response to blood volume; θ_4_, general curve parameter; ϕ_1_, first-order model drug elimination subject-specific half-life for sodium nitroprusside; ϕ_2_, onset time (including infusion delay, circulation time, and drug/target properties) for sodium nitroprusside; ϕ_3_, curve effect of sodium nitroprusside on mean arterial pressure; ϕ_4_, gain effect of sodium nitroprusside on mean arterial pressure; ϕ_5_, curve effect of sodium nitroprusside on venous tone and cardiac stroke volume; ϕ_6_, gain effect of sodium nitroprusside on venous tone and cardiac stroke volume.

**Table 2 jpm-13-01101-t002:** Model errors for hemodynamics by fit type.

	Individually Fit Models			Population Models			Partially Fit Models		
Subject	SV	CO	MAP	SV	CO	MAP	SV	CO	MAP
1	9.6	0.8	5.5	13.2	1.1	9.5	10.7	0.9	5.4
2	4.7	0.4	3.7	26.9	1.9	7.8	9.3	0.7	4.4
3	7.5	0.6	5.2	13.3	0.9	9.3	7.9	0.6	4.3
4	7.4	0.6	4.5	8.6	0.7	8.4	7.4	0.6	4.6
5	12.1	1.0	8.6	15.0	1.3	20.4	13.7	1.2	9.4
6	15.1	1.3	6.5	16.3	1.3	11.9	16.8	1.3	6.5
7	18.6	1.6	4.4	17.5	1.7	9.3	16.5	1.5	4.9
Mean	10.7	0.9	5.5	15.8	1.3	11.0	11.8	1.0	5.7
SD	4.9	0.4	1.6	5.7	0.4	4.4	3.9	0.4	1.8

Error values are the root-mean-squared errors between the model-predicted values and the observed values. Individually fit models were markedly superior to using the mean population values. Partially fit models were generally comparable to fully individualized models.

## Data Availability

The data presented in this study are available on request from the corresponding author.

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
