# Peer review of "Physiological Modeling of Hemodynamic Responses to Sodium Nitroprusside"

_jpm, 2023, doi:10.3390/jpm13071101_

Round 1
Reviewer 1 Report
The authors have further refined a previously developed computational model predictive of various hemodynamic parameters. The project used in-silico data from a swine model in which hypotension was induced by sodium nitroprusside infusion. The computational model was improved by adjusting the variables to attain a decrease in the RSME. With the improved model, a comparison was made visually superimposing the actual individual experiment data in stroke volume, cardiac output, and mean arterial pressure with the simulated data and subsequently reporting Bland Altman Plots demonstrating good agreement between the datasets. The work lines up appropriately with the growing efforts to develop automated closed-loop systems for clinical management, e.g., during surgery and in the ICU.
Reviewer 2 Report
This manuscript proposes and calibrates lumped-parameter computational models of cardiovascular physiology to account for the effects of Sodium Nitroprusside (SNP) on blood pressure and blood volume. Existing models for the cardiovascular system, including blood volume changes, were extended by adding mechanisms through which SNP would theoretically act. Animal experiments were performed to analyse the sensitivity of several model parameters and to find subject-specific values that gave the best fit to experimental measurements of mean arterial pressure and stroke volume.
Specific remarks:
1. The Introduction should provide more context into potential applications of cardiovascular models including the effect of SNP. The overall aims of why the model is developed in the first place are not clear enough.
2. Page 3, equations (5),(12),(14): How are u(t) and v(t) defined? Are they model inputs or state variables to be solved? Clarify.
3. Page 3: There is no equation for the effect of SNP including the parameters phi_1 and phi_2. Please add details on this modelling term.
4. Section 2.3: Provide more details on the "advanced hemodynamic monitoring" system.
5. Section 2.4: Were information theoretical measures (information criteria etc.) used to estimate the utility of adding this many parameters?
6. Table 1: The subject-specific parameter fits have quite a lot of variance. Can the authors comment on the credibility of the models of blood pressure regulation when the parameters vary up to 100%? This could be an indication of improper model structure or missing terms.
7. Page 6: The grid-search method seems ad hoc and arbitrary. Why 10% increase/decrease in each parameter? Was this before or after normalisation? Why were different parameters calibrated in sets and why in this specific order? Together with the large variance of subject-specific parameter values this raises some concerns about the validity of the grid-search approach.
8. Table 2: Can the authors comment on the inability of the models to accurately match SV values? If I understood, equation (12b) should be used to compute the model-estimated SV. Is this an indication that the structure of the equation (12b) may need to be improved?
9. Figure 1: The authors mentioned that BP values for some of the pigs (case 7) caused the model to not fit the experiments very well. Can the authors add the BP into this plot somehow? It would aid in understanding why the fit is so much better in some cases than others.
This work is very thorough and explores systematically the models it develops and performs high quality experimental calibrations to their parameter sets. The experimental fit is quite good (although does not fully work in all cases). My main doubts are about the general utility of the model, whether the chosen model structure is appropriate, and a wish to see more justification for the usefulness of the model. I recommend major revision to address the point above.
The manuscript is very clearly written, with only a few typographic problems.
Round 2
Reviewer 2 Report
The authors have addressed my remarks and the paper is now of sufficient quality to be accepted.
The quality of the writing is good, it should still be checked for typos.